# Distinct mRNAs in Cancer Extracellular Vesicles Activate Angiogenesis and Alter Transcriptome of Vascular Endothelial Cells

**DOI:** 10.3390/cancers13092009

**Published:** 2021-04-22

**Authors:** Pan Zhang, Su Bin Lim, Kuan Jiang, Ti Weng Chew, Boon Chuan Low, Chwee Teck Lim

**Affiliations:** 1NUS Graduate School—Integrative Sciences and Engineering Programme (ISEP), National University of Singapore, Singapore 119077, Singapore; zhangp@u.nus.edu; 2Department of Biomedical Engineering, National University of Singapore, Singapore 117583, Singapore; sblim@ajou.ac.kr; 3Department of Biochemistry and Molecular Biology, Ajou University School of Medicine, Suwon 16499, Korea; 4Mechanobiology Institute, National University of Singapore, Singapore 117411, Singapore; jiangkuan@u.nus.edu (K.J.); mbichew@nus.edu.sg (T.W.C.); 5Department of Biological Sciences, National University of Singapore, Singapore 117558, Singapore; 6University Scholars Programme, National University of Singapore, Singapore 138593, Singapore; 7Institute for Health Innovation and Technology (iHealthtech), National University of Singapore, Singapore 117599, Singapore

**Keywords:** extracellular vesicles, messenger RNA, cancer biomarker, hypoxia, tumor microenvironment, angiogenesis

## Abstract

**Simple Summary:**

Cancer extracellular vesicles (EVs) are implicated in various processes of cancer development, with most of the EV-induced changes attributed to EV proteins and microRNAs. However, the knowledge about the cancer EV-mRNAs remains limited. Here, we have assessed the mRNAs of 61 diverse oncogenes and found half of them, including *VEGFA* and *SNAIL1/2*, are abundant in cancer EVs while absent in non-tumorigenic cell-derived EVs. Fluorescent trafficking shows the EV VEGFA mRNAs are translatable after being internalized by the recipient cell. Concomitantly, the cancer EVs induced VEGFA-dependent angiogenesis and upregulated epithelial-mesenchymal transition-related genes. Our findings reveal that the EV-mRNA profile can reflect the cell malignancy, and the intercellular transfer of these mRNAs can contribute toward tumor angiogenesis.

**Abstract:**

Cancer-derived extracellular vesicles (EVs) have been demonstrated to be implicated in various processes of cancer development, with most of the EV-induced changes attributed to EV-proteins and EV-microRNAs. However, the knowledge about the abundance of cancer EV-mRNAs and their contribution to cancer development remain elusive. Here, we show that mRNAs prevail in cancer EVs as compared with normal EVs, and cancer EVs that carry abundant angiogenic mRNAs activate angiogenesis in human umbilical vein endothelial cells (HUVECs). Specifically, of a gene panel comprising 61 hypoxia-targeted oncogenes, a larger proportion is harbored by cancer EVs (>40%) than normal EVs (14.8%). Fluorescent trafficking indicates cancer EVs deliver translatable mRNAs such as *VEGFA* to HUVECs, contributing to the activation of VEGFR-dependent angiogenesis and the upregulation of epithelial-mesenchymal transition-related and metabolism-related genes. Overall, our findings provide novel insights into EV-mRNAs and their role in angiogenesis, and has potential for diagnostic and therapeutic applications.

## 1. Introduction

Extracellular vesicles (EVs) are cell-derived lipid bilayer vesicles that carry proteins and nucleic acids. Though the mechanism of loading molecules into EVs is not fully understood yet, the EVs, including small EVs (sEVs) with a diameter of approximately 100 nm, are known to deliver various functional proteins and nucleic acids, especially microRNAs (miRNAs), to recipient cells, which plays a critical role in intercellular communications [1,2,3,4,5,6]. Particularly, cancer cell-derived EVs are known to advance cancer progression by suppressing the immune response, educating the neighboring cells, and initiating the pre-metastatic niche formation [7]. As EVs are circulating in the body fluid, they serve as promising non-invasive surrogate markers for tumor tissue, in which repeated sampling is not clinically practical given their invasive nature [8,9]. Furthermore, EVs are detected more frequently in plasma or serum (10^9^–10^12^/mL) and are more stable, compared to circulating tumor cells (CTCs) and cell-free DNA (cfDNA), respectively, highlighting their potential applications in a clinical setting [10]. Despite such progress, the lack of reliable markers for tumor-derived EVs calls for comprehensive profiling and the development of new technologies enabling identification of selective subtypes of EVs [11,12].

Along with the growing understanding of EV cargo, the EV-proteins [13,14], non-coding RNAs [15] and mutated nucleic acids [16] have also been proposed to be cancer biomarkers. However, the EV-mRNAs have rarely been investigated. Prior evidence demonstrated the presence of several hypoxia-targeted mRNAs in the cancer EVs [17]. Hypoxia is a common signature of the tumor microenvironment (TME) attributed to the insufficient supply of blood to the intensively proliferating tumor. The hypoxic stress could regulate the transcription of target genes though hypoxia-inducible factors (HIFs) and thus contribute to tumor progression [18]. Recently, hypoxia-induced EVs have been found to mediate angiogenesis, metastasis and immunosuppression [19]. These effects were attributed to EV-protein and miRNAs, while the contributions of EV-mRNAs remain elusive. Here, we aimed to comparatively characterize the EVs derived from cancer cells and non-tumorigenic cells and assess the potential role of cancer EV-mRNAs in inducing VEGFR-dependent angiogenesis in vitro and transcriptome changes in human umbilical vein endothelial cells (HUVECs).

## 2. Materials and Methods

### 2.1. Cell Culture

MCF10A, MCF7, MDA-MB-231, Caco2, SW480, SW620, MIAPaCa-2 and HepG2 were purchased from American Type Culture Collection (ATCC, Manassas, VA, USA). H1650 and H1792 were obtained from National Cancer Center Singapore. GC38 was a gift from Dr. Shing-Leng Chan from the Cancer Science Institute of Singapore. As for the culture medium, the cells MCF7, MDA-MB-231, SW480, SW620, MIAPaCa-2 and GC38 were cultured in DMEM (Gibco; Thermo Fisher Scientific, Waltham, MA, USA, Cat No. 11965092) supplemented with 10% fetal bovine serum (FBS; Gibco; Thermo Fisher Scientific, Cat No. 26140079). Caco2 was cultured in DMEM supplemented with 20% FBS. HepG2 was cultured in EMEM (Gibco; Thermo Fisher Scientific, Cat No. 11095080) supplemented with 10% FBS. H1650 and H1792 were cultured in RPMI-1640 (Gibco; Thermo Fisher Scientific, Cat No. 11875093) supplemented with 10% FBS. MCF10A, MCF7, MDA-MB-231 and HUVECs were cultured for up to 10 passages (7 passages for HUVECs). The mycoplasma was tested regularly (every 6 months) by MycoAlert Mycoplasma Detection Kit (Lonza, Basel, Switzerland, Cat No. LT07-418) and was confirmed negative each time. These cells were cultured under 20% O_2_ for normoxia and 1% O_2_ for hypoxia.

### 2.2. EV Incubation and Isolation

EVs were harvested following the Minimal Information for Studies of Extracellular Vesicles (MISEV) guidelines [20]. Cells were cultured in complete medium under normoxia conditions till up to 75–90% confluency in a T75 cell culture flask. The culture medium was removed, and the cells were washed three times with PBS. Fresh serum-free basal medium with high glucose was added to incubate the cells. The cell culture flasks were put in either a normoxia (20% O_2_) or hypoxia (1% O_2_) incubator for 24 h for the cells to adapt to these oxygen conditions. Next, the medium was replaced with fresh serum-free medium, and the cells were incubated in the same incubator for another 24 h to produce either normoxic or hypoxic sEVs. The supernatant of the cell culture was collected by pipetting. The sEVs in culture medium were enriched and purified by differential centrifugation following the established protocol [21]. Specifically, cell debris was removed by 10 min of 2000× *g* centrifugation. Larger vesicles including apoptotic bodies and microvesicles were excluded by 30 min of 10,000× *g* centrifugation. Finally, sEVs were pelleted by 2 h of 100,000× *g* centrifugation with the Fiberlite F50L-24 × 1.5 Fixed-Angle Rotor (Thermo Finisher Scientific). The pellet was then resuspended in 1mL PBS and centrifuged again. The enriched EVs were kept at 4 °C and processed within 3 days, or stored under −80 °C without cryoprotectants for longer term storage and used within 15 days.

### 2.3. Quantitative Analysis of EVs by Nanoparticle Tracking Analysis

Varied numbers of MCF10A, MCF7 and MDA-MB-231 cells were cultured in T25 flasks. Prior to the sEV incubation, cells were starved in serum-free high-glucose medium for 3–6 h. Next, the flasks were washed by 5 mL PBS three times, and 2 mL new serum-free medium was added into the flask to incubate the cells for sEV production. After 3 h incubation, the culture supernatant was collected by pipetting. The larger extracellular vesicles in the medium were removed by 30 min 10,000× *g* centrifugation. The remaining medium containing sEVs was diluted properly and analyzed by the NanoSight LM10 (Malvern Panalytical, Malvern, UK) with the following parameters: camera level = 13; detect threshold = 3; max jump distance = auto; filming time for each video clip was 10 s; and each sample was filmed at 10 different points. Meanwhile, the cells in the T25 flask were harvested and counted by the Multisizer 4e Coulter Counter (Beckman Coulter, Brea, CA, USA).

The accuracy of the quantitative analysis was optimized with the following measures: first, the EV incubation was constrained to a short 3-h period that was much shorter than the cell doubling time (more than 16 h) so that the total cell number was minimally affected by cell proliferation; second, the samples with discrepant concentrations were diluted accordingly to achieve similar final concentrations, thus the comparison of the concentrations between samples became reliable independent of the technical linear range of the NTA; third, the data of each sample was acquired by averaging the measurements of 10 random regions, reducing the effects of the heterogeneity among regions.

### 2.4. PKH26 Staining of Cells and EVs

Confluent cells were trypsinized and stained by PKH26 Red Fluorescent Cell Linker Kit (Sigma-Aldrich, St. Louis, MO, USA, Cat No. PKH26GL) following the vendor’s instruction. Briefly, less than 1 × 10^7^ cells were pelleted and resuspended in 1 mL Diluent C, and then mixed with 1 mL 4 × 10^−6^ M PKH26 for 5 min. Next, the staining was stopped by incubating with 2 mL fetal bovine serum for 1 min. Then, the stained cells were washed with complement culture medium three times to remove excess dye. Last, the cells were seeded in flasks to produce EVs, or in petri dishes for fluorescent imaging.

To stain the EVs, we refrained from directly mixing the PKH26 dye with EVs because the excess dye co-isolated with EVs and mimicked EVs even in the control PBS sample (Appendix A). The issue of contamination by excess PKH dye was also reported in other studies [22,23]. Instead, the PKH26 dye was used to only stain the cells, followed by a triple wash with complete culture medium to thoroughly remove the excess dye. Taking the advantages that the PKH26 staining is efficient and the fluorescence lasts for weeks, the EVs generated by the PKH26^+^ cells were also positive for PKH26 (Appendix A).

### 2.5. Western Blot

The proteins of cells and EVs were extracted with the RIPA as described in the previous study [24]. The total protein levels of the lysates were measured by BCA Protein Assay Kit (iNtRON Biotechnology, Seongnam-si, South Korea, Cat No. 21071). The following primary antibodies were used at 1:1000 dilution: Alix (Cell Signaling Technology, Danvers, MA, USA, Cat No. 2171), CD63 (Abcam, Cambridge, UK, Cat No. ab59479), CD9 (Abcam, Cat No. ab92726), CD81 (Abcam, Cat No. ab79559), VEGFA (Abcam, Cat No. ab214424), VEGFR1 (Abcam, Cat No. ab32152), VEGFR2 (Abcam, Cat No. ab134191), Erk1/2 (Cell Signaling Technology, Cat No. 4695) and β-actin (Invitrogen; Thermo Fisher Scientific, Cat No. MA5-15739-HRP). The primary antibody phospho-Erk1/2 (Sigma-Aldrich, Cat No. M9692) was used at 1:3000 dilution. The secondary antibodies were used at 1:3000 dilution. The intensity of the bands was analyzed by the ImageJ “gel analysis” function. Briefly, multiple bands were selected by identical squares, and the pixel intensity distribution along the y-axis was plotted for each square. Area under the distribution curve was analyzed and normalized by rows so that the sum in each row was consistently 10.

### 2.6. RT-qPCR and DNA Gel Electrophoresis

Total cell RNA was extracted by RNeasy Kit (Qiagen, Hilden, Germany, Cat No. 74106). Total EV RNA was isolated by Total EV RNA & Protein Isolation Kit (Invitrogen; Thermo Fisher Scientific, Cat No. 4478545). RNA was reverse transcribed to cDNA by SuperScript III Reverse Transcriptase (Invitrogen; Thermo Fisher Scientific, Cat No. 18080044), followed by RT-qPCR with the FastStart Universal SYBR Green Master (Sigma-Aldrich, Cat No. 4913850001). Primer sequences are listed in Appendix A.

Two percent agarose gel was prepared by dissolving 2 g Agarose (Bio-rad, Cat No. 1613102) and 2 µL SYBR Safe DNA Gel Stain (Invitrogen; Thermo Fisher Scientific, Cat No. S33102) in 100 mL Tris-acetate-EDTA (TAE) buffer. Amplified DNA of target genes was mixed with 6× DNA gel loading dye (Thermo Fisher Scientific, Cat No. R0611) and subjected to gel electrophoresis at 105 V for 30 min. GeneRuler 100 bp DNA Ladder (Thermo Fisher Scientific, Cat No. SM0241) was used as the reference for DNA size. Imaging of the gel was done by iBright FL1000 (Invitrogen; Thermo Fisher Scientific).

### 2.7. Fluorescent Imaging of Cells and EVs

Cells were fixed on glass by 4% paraformaldehyde solution in PBS. Hoechst 33342 (Thermo Fisher Scientific, Cat No. 62249) was used at 1:1000 dilution to stain cell nuclei. The cell actin was stained by Phalloidin-iFluor 647 (Abcam, Cat No. ab176759) at 1:2000 dilution for 30 min. After the staining, the fixed samples were imaged by Nikon A1R confocal microscopy.

EVs were sealed on glass slides for imaging. Briefly, 10 µL of EVs were dripped onto the glass slide. A coverslip was placed on top of the drop to spread the liquid. Next, nail polish was applied to immobilize and seal the coverslip. The slide with the EVs was imaged by Nikon A1R confocal microscopy with the coverslip side facing the lens.

### 2.8. Co-Culture and Fluorescent Trafficking of EVs and mRNAs

CD63-GFP was a gift from Paul Luzio (Addgene, Watertown, MA, USA, plasmid # 62964). MDA-MB-231 cells were transfected with the CD63-GFP by Lipofectamine 3000 (Invitrogen; Thermo Fisher Scientific, Cat No. L3000015). The CD63-GFP^+^ MDA-MB-231 cells were selected by gentamicin and enriched by flow cytometry. One day before co-culture, 1 × 10^4^ HUVECs were seeded in glass-bottom 8-well µ-slides and incubated overnight. After the HUVECs adhered, 1 × 10^4^ GFP^+^ MDA-MB-231 were harvested, stained with PKH26 and added to the HUVECs. After a 24-h co-culture in 1:1 mixed EGM-2 and FBS-supplemented DMEM, the cells were fixed and imaged at 100× magnification by the confocal Nikon A1R (Tokyo, Japan).

SYTO RNASelect Green Fluorescent Cell Stain (Thermo Fisher Scientific, Cat No. S32703) was used to stain the RNA in MDA-MB-231 at 1:10,000 dilution. Approximately 1 × 10^5^ PKH26^+^ SYTO RNASelect^+^ MDA-MB-231 cells were first seeded in a 35 mm glass-bottom petri dish. Excess dye was removed by three-times PBS wash. Next, 1 × 10^5^ HUVECs were added to the petri dish and imaged by confocal microscopy with a 60× magnification lens at 1-h intervals.

The VEGFA-GFP plasmid was constructed by the protein cloning Core of Mechanobiology Institute in Singapore. The GFP were fused with VEGF165 at the c-terminus of VEGF165. MDA-MB-231 cells were first stained with PKH26, followed by transient transfection of the VEGFA-GFP plasmid with Lipofectamine 3000 (Thermo Fisher Scientific, Cat No. L3000015). The next day, the transfected cells were incubated in serum-free DMEM for EV generation. After 24 h, the culture medium was collected, and the EVs in the culture medium were enriched and purified. These EVs (50 µg/mL) were then used to treat the HUVECs for 24 h. Last, the HUVECs were fixed and imaged with 100× magnification.

### 2.9. Tube Formation Assay

The 24-well plate was coated with 80 µL growth factor-reduced Matrigel (Bio Laboratories, Cat No. 354263). After gel formation, 1 × 10^5^ HUVECs were seeded in each well. The cells were incubated in EBM-2 basal medium for control, treated with 50 µg/mL MDA-MB-231-HEVs or treated with both 50 µg/mL MDA-MB-231-HEVs and 0.1 µM Axitinib (Selleck Chemicals, Houston, TX, USA, Cat No. S1005). After 24 h, the cells were stained with Calcein AM (Thermo Fisher Scientific, Cat No. L3224), followed by imaging of multiple positions under 10× magnification. Tube length in each image was analyzed by the Angiogenesis Analyzer plugin for ImageJ [25].

### 2.10. EdU Proliferation Assay

The Click-iT™ EdU Proliferation Assay Kit (Invitrogen; Thermo Fisher Scientific, Cat No. C10499) was used for the proliferation assay of HUVECs following the manufacturer’s instruction. Briefly, HUVECs were seeded in 24 wells, plated and incubated overnight. The adhered HUVECs were starved in EBM-2 for 3 h, followed by incubation with 10 µM EdU in either basal EBM-2 or 50 µg/mL MDA-MB-231-HEVs. After 24 h, the HUVECs were fixed by Click-iT EdU fixative. Then, the cell nucleus was stained by Hoechst 33342. By fluorescent imaging, the total cells were counted with nucleus fluorescence and the proliferative cells were counted with EdU fluorescence.

### 2.11. RNA-Seq and Data Analysis

HUVECs were incubated in basal EBM-2 for control or treated with 50 µg/mL MDA-MB-231-HEVs for 24 h. The total cell RNA of two control samples and two treated samples were extracted with RNeasy Kit (Qiagen, Cat No. 74106), and then processed and sequenced by BGI Genomics (BGI Genomics, Shenzhen, China).

With the RNA-seq data, the sequencing quality were verified and the correlations between samples were analyzed by BGI Genomics. The filtered clean reads were mapped to the human genome reference GRCh38.p13 Ensembl release 97 by two parallel pipelines, HISAT2-HTSeq [26,27] and Salmon [28]. Based on the gene counts acquired by the two algorithms, differentially expressed genes (DEGs) were analyzed by the two competing approaches, DESeq2 [29] and edgeR [30] with the thresholds of 0.05 adjusted *P*-value or 0.1 false discovery rate, respectively. Consequently, four lists of DEGs were acquired and a final DEG list was obtained by intersecting the four lists. The gene ontology enrichment analysis of the DEGs was done with The Gene Ontology Resource (http://geneontology.org/, accessed on 24 March 2020), and the gene set enrichment analysis (GSEA) was finished with the GSEA software (www.gsea-msigdb.org/, accessed on 9 March 2020) [31].

### 2.12. Statistical Analysis

The comparison between two sets of data was evaluated by using two statistical models: Student’s *t* test for normally distributed data and Mann–Whitney U test for nonparametric data. For Student’s *t* test, two-tailed significance was reported with an equal variance assumption. The computation was completed by Excel (Microsoft) and R [32]. The specific tests used are specified in every figure legend.

## 3. Results

### 3.1. Cancer Cells Secreted EVs at Prominently Higher Rates Than Normal Cells

EVs were produced by the incubation of three breast cell lines in serum-free medium—cancer cell MCF7 and MDA-MB-231 (MDA231), as well as non-tumorigenic cell MCF10A (Figure 1A, left panel). With the lipophilic dye PKH26 that uniformly and thoroughly stained the cytoplasmic membranes of the cells, the EVs isolated from the culture medium of the PKH26^+^ cells were also observed to have the PKH26 fluorescence (Figure 1A, right panel). By western blotting, the enriched EV-like vesicles from the culture medium were confirmed positive for EV protein markers including Alix, CD63, CD9 and CD81 (Figure 1B). With larger vesicles in the culture medium removed by 10,000× *g* centrifugation for 30 min, the remaining vesicles were characterized by Nanoparticle Tracking Analysis (NTA) (Figure 1C). The size of these vesicles ranged from 30 to 150 nm, which was consistent with that of sEVs [2] (Figure 1D, upper panel). To exclude the possibility of contamination with very low density lipoprotein (VLDL), which overlaps with sEVs in size and density [33,34,35,36], we distinguished sEVs (1.1–1.2 g/mL) and VLDL (~1.006 g/mL) through centrifugation. We demonstrated that the observed vesicles could be completely removed from the supernatant by 100,000× *g* ultracentrifugation for 2 h, excluding the possibility that they might be lipoproteins (Figure 1C). These results provide evidence that the vesicles obtained and analyzed were sEVs.

The normalized size distribution of EVs was averaged from six samples for MCF10A, 11 samples for MCF7 and eight samples for MDA231. Notably, the curve for MDA231-derived EVs (MDA231-EVs) showed a shift towards a larger size compared to MCF10A-derived EVs (MCF10A-EVs) and MCF7-derived EVs (MCF7-EVs) (Figure 1D, upper panel). Dividing the sEV population into the small (sEV-S) and the large (sEV-L) subtypes by an 85 nm size cut-off according to a previous study [37], the proportion of the sEV-L population for MDA231-EVs (67.4% ± 3.7%) was substantially higher than MCF7 (53.9% ± 3.6%) and MCF10A (50.9% ± 4.4%) (Figure 1D, lower panel). These results indicate that the metastatic cancer cells of MDA231 tend to generate larger EVs than MCF10A and MCF7 cells.

To quantitatively evaluate the EV generation rate of each cell line, we engaged a coulter counter and NTA to count the cells and the secreted EVs, respectively. Multiple measures were implemented to enhance the accuracy of the measurements (more details in Methods). Importantly, we noticed that the cell generated EVs at a constant rate independent of the cell density, since the EV number linearly correlated with the cell number. By linear regression with a zero intercept, the EV generation rates were calculated to be 0.96 × 10^9^ (95% confidence interval (CI) = 0.82 × 10^9^, 1.09 × 10^9^) per million cells per hour for MCF10A, 2.53 × 10^9^ (95% CI = 2.19 × 10^9^, 2.86 × 10^9^) per million cells per hour for MCF7 and 6.27 × 10^9^ (95% CI = 5.22 × 10^9^, 7.33 × 10^9^) per million cells per hour for MDA231 (Figure 1E). Therefore, the EV generation rate is cell line-dependent, and breast cancer cells secrete EVs at a folds higher rate than healthy cells.

### 3.2. EV-mRNA Profile Showed Low Correlation with That of Cells

We evaluated the presence and abundance of mRNAs in EVs with RT-qPCR. The housekeeping gene GAPDH was first verified to be ubiquitously present in EVs (Appendix A), and thus was used as an endogenous reference to normalize other mRNA levels. We assessed the EV-mRNAs of a diverse panel comprising 61 hypoxia-targeted oncogenes (Appendix A, Dataset S1; hereinafter referred to as the hypoxia gene panel) which were reported to be transactivated by hypoxia-inducible factors (HIFs) and can contribute to tumor progression [38,39]. The low coefficient of variance (<0.3) of the detected mRNA levels suggested that the EV-mRNA levels were intrinsically stable and the quantification by RT-qPCR was reproducible (Appendix A).

We first compared the mRNA profile between EVs and their original cells. By RT-qPCR, the cellular expressions of a proportion of the panel were assessed (Appendix A), and the inconsistency between EV and cell were noticed. For example, the genes *MET*, *P4HA2*, *ANGPTL4* and *VEGFA* showed higher expression levels in MCF10A cells than MDA231 cells, but these transcripts were not loaded into MCF10A EVs (Appendix A). Furthermore, the hypoxic stress downregulated MET expression in cells but upregulated MET mRNA abundance in EVs (Appendix A). We also analyzed the cell mRNA profile from an online database (GSE75168, GSE111653) and found the correlation between cell and EV-mRNA level was weak, with the R^2^ value ranging from 0.2878 to 0.6364 (Appendix A). Moreover, the hypoxic stress induced a contrary change of mRNA levels in cells and EVs for 38–48% of the regulated genes (Appendix A). Taken together, these inconsistencies between the EVs and cells demonstrate that the mRNA profile of EVs is independent of that of cells.

### 3.3. Cancer EVs Harbored Diverse Hypoxia-Targeted mRNAs

Comparing the EV-mRNA profile across the cell lines, we found the cancer EVs carried a substantially larger proportion of the hypoxia panel than normal EVs under either normoxia or hypoxia (Figure 2A). Overall, 14.8% (9/61) of mRNAs were detected in normoxic MCF10A-EVs (MCF10A-NEVs), 42.6% (26/61) of mRNAs in normoxic MCF7-EVs (MCF7-NEVs), 49.2% (30/61) of mRNAs in normoxic MDA231-EVs (MDA231-NEVs), 19.7% (12/61) of mRNAs in hypoxic MCF10A-EVs (MCF10A-HEVs), 50.8% (31/61) of mRNAs in hypoxic MCF7-EVs (MCF7-HEVs) and 60.7% (37/61) mRNAs in hypoxic MDA231-EVs (MDA231-HEVs) (Figure 2B). Altogether, a total of 49.2% (30/61) of mRNAs were found to be present in normoxic cancer EVs while absent in MCF10A-NEVs, highlighting their potential as cancer biomarkers (Appendix A). Of these 30 genes, 26.7% (8/30; *CXCL12*, *AMF*, *SNAIL1*, *TAZ*, *PHGDH*, *PDK1*, *MXI1* and *SIAH2*) showed higher levels in MCF7-NEVs and the other 73.3% (22/30; *VEGFA*, *LOX*, *P4HA1*, *P4HA2*, *MET*, *SNAIL2*, *ROCK1*, *RHOA*, *AXL*, *PLAUR*, *ANGPTL4*, *MMP14*, *IL8*, *CD47*, *IL6*, *SOX2*, *GLUT1*, *PLIN2*, *BNIP3L*, *SHMT2*, *LONP1* and *SLC16A4*) showed higher levels in MDA231-NEVs (Appendix A).

### 3.4. Hypoxia Regulated Cancer EV-mRNAs in a Cell Line-Dependent Manner

Next, we analyzed the hypoxia-induced alterations of the EV-mRNAs. The hypoxic stress induced additional mRNAs to be loaded into EVs by MCF10A (*N* = 4), MCF7 (*N* = 5) and MDA231 (*N* = 7). Six of these mRNAs (*LOXL2*, *ZEB2*, *IL19*, *CA9*, *FABP3* and *NT5E*) were absent in all normoxic EVs (Appendix A), which are potential hypoxia biomarkers that can be used to reflect the hypoxic status in the TME. Notably, MCF7-HEVs and MDA231-HEVs harbored 26 common mRNAs, all of which exhibited > 2 folds higher levels in the MDA231-HEVs than the MCF7-HEVs (Figure 2C). Furthermore, we noticed that MCF7 and MDA231 responded to hypoxia in an opposite manner in terms of the EV-mRNA level. Of the significantly altered EV-mRNAs, 92.3% (12/13) were downregulated for MCF7, while 75% (12/16) were upregulated in MDA231 (Appendix A). These findings suggest that in the loading of mRNAs into EVs, the hypoxic stress may be suppressive for benign cancer cells while favorable for aggressive cancer cells, highlighting the potential of EV-mRNAs in distinguishing tumor stages.

We verified the ubiquity of the EV-mRNAs derived from multiple cancer cell lines with the VEGFA mRNA as a candidate for proof-of-principle. Importantly, VEGFA mRNA was detected in EVs from all eight cancer cell lines tested, including colorectal cancer cells (CACO-2, SW480, SW620), pancreatic cancer cells (MiaPaCa-2), lung cancer cells (H1650, H1792), gastric cancer cells (GC38) and liver cancer cells (HepG2) (Figure 2D). Of all the tested cancer cell lines, 70% (7/10) showed significant changes in the EV-VEGFA mRNA level induced by hypoxia. Interestingly, the VEGFA protein levels in EVs and cell lysates were not concomitantly modulated by hypoxia (Appendix A), indicating that the hypoxia mediated independent impacts on the loading of mRNAs and proteins into EVs.

### 3.5. Cancer EVs Transferred RNAs to HUVECs

To test the hypothesis that cancer EVs deliver mRNAs to recipient cells, we co-cultured MDA231 cells and HUVECs (Figure 3A). To first observe the uptake of cancer EVs by HUVECs, the CD63-GFP^+^ MDA231 cell line was established and stained with PKH26 before co-culture with HUVECs, so that cell-derived EVs could be labeled either by CD63 or PKH26 (Appendix A). After a 24-h co-culture of CD63-GFP^+^, PKH26^+^, MDA231 and HUVECs, both PKH26 and CD63 were spotted inside some of the HUVECs with some of their spots overlapping, demonstrating that the CD63^+^ EVs or the PKH26^+^ EVs derived from MDA231 were internalized by HUVECs within a period of 24 h (Figure 3B,C). To further demonstrate the transfer of RNA from MDA231 to HUVECs, the two cell lines were co-cultured, with the RNAs of the PKH26^+^ MDA231 stained by the SYTO RNAselect green fluorescent dye (Figure 3D). The RNAselect dye was verified to specifically stain RNAs and can last for more than 24 h (Appendix A). Staining the PKH26^+^ MDA231-EVs with the RNAselect dye confirmed the presence of RNAs in some of the EVs (Appendix A). Live cell imaging of HUVECs co-cultured with the SYTO RNAselect^+^ PKH26^+^ MDA231 cells showed a gradual accumulation of fluorescent RNAs in HUVECs within 15 h (Figure 3E,F). These results demonstrate that MDA231 cells transferred RNAs to HUVECs via EVs.

### 3.6. Recipient HUVECs Translated Cancer EV-VEGFA mRNA

To show that translatable mRNAs can be delivered by EVs, we tagged the VEGFA with GFP and tracked the transfer of VEGFA-GFP from MDA231 to HUVECs through EVs (Figure 4A). The VEGFA-GFP plasmid was constructed with the fusion of GFP at the c-terminus of VEGF-165 so that the biological activity of VEGFA remained undisturbed [40]. With transient transfection, MDA231 cells expressed VEGFA-GFP and loaded an appreciable amount of VEGFA-GFP mRNA into EVs (Figure 4B,E). These EVs were then used to treat HUVECs. Prior studies used 10–100 µg/mL EVs in co-culture experiments [17,41,42]. Therefore, we used fixed EV concentrations at an intermediate value of 50 µg/mL. This EVs concentration was not unreasonably high considering that the in vivo circulating EV concentration is 500–1500 µg/mL in cancer patient [15]. With 24-h incubation, the EV-VEGFA-GFP mRNA was found to be internalized by HUVECs (Figure 4E). Fluorescent imaging showed that VEGFA-GFP protein and the PKH26 dye were present in all HUVECs (*N* > 100) (Figure 4C). Since no detectable level of VEGFA-GFP was observed in the EVs (Figure 4D), we conclude that the VEGFA-GFP protein in HUVECs was synthesized through translating the EV-transferred VEGFA-GFP mRNA. These findings indicate that EV-mRNAs can be translatable in recipient cells.

### 3.7. Cancer EVs Activated VEGFR-Dependent Angiogenesis in HUVECs

The hypoxia-regulated VEGFA has been recognized as a major driving force for angiogenesis [43]. We asked if the cancer EVs that carried abundant VEGFA mRNA could activate angiogenesis. The tube formation assay of the HUVECs revealed that MDA231-HEVs (50 µg/mL) substantially increased the total length of tubes with 24-h incubation. This increase was completely halted by the VEGF receptor inhibitor Axitinib (0.1 µM) (Figure 5A,B). Using RT-qPCR, expressions of VEGFA, VEGFR1 and VEGFR2, which are the key genes for the VEGFR-dependent angiogenesis pathway, were found to be significantly upregulated in the HUVECs treated with the MDA231-HEVs. On the other hand, the ANGPT-TIE angiogenesis pathway-related key genes, including ANGPT1, ANGPT2, TIE1 and TIE2, remained unaffected (Figure 5C). Moreover, western blotting confirmed the enhanced phosphorylation of Erk1/2, which is downstream of the angiogenic signaling transduction, as well as the elevation of the VEGFR1 protein (Figure 5D). Taken together, these data suggest that cancer EVs carrying abundant VEGFA mRNAs can activate the VEGFA-VEGFR angiogenesis in endothelial vascular cells.

The activation of the VEGFA-VEGFR angiogenesis pathway is known to promote various cellular activities such as cell proliferation, survival and migration [44]. However, an EdU proliferation assay showed that the proliferation rate of HUVECs was reduced by the MDA231-HEVs (50 µg/mL) from 31.1 ± 0.83 to 17.4 ± 0.68% (Appendix A). To further verify this EV-induced inhibition of cell proliferation, we assessed the expression of cell cycle-related genes, including six cyclins (CCNA1, CCNA2, CCND1, CCND2, CCND3 and CCNE2), five cyclin-dependent kinase (CDK1, CDK2, CDK4, CDK6 and CDK7) and eight cyclin-dependent kinase inhibitors (CDKN1A, CDKN1B, CDKN1C, CDKN2A, CDKN2B, CDKN2C, CDKN2D and CDKN3) in HUVECs. Of all significantly altered genes, two pro-proliferative genes (CCNA2 and CDK1) were downregulated, and the two anti-proliferative genes (CDKN1A and CDKN1B) were upregulated by MDA231-HEVs (Appendix A), supporting the observed attenuation of cell proliferation. These data suggest that the MDA231-HEVs-induced enhancement of tube formation was not accompanied by augments in cell proliferation. Notably, the mRNAs of the two upregulated genes, CDKN1A and CDKN1B, were carried by the MDA231-HEVs, indicating that the EV-mediated transfer of anti-proliferative mRNAs may contribute to the suppression of cell proliferation (Appendix A).

### 3.8. Cancer EV-Mediated Alterations in HUVEC Transcriptome Overrepresented Cell Motility and Metabolism

The MDA231-HEVs-mediated changes in the HUVEC transcriptome were investigated using RNA sequencing. Two control samples and two EV-treated samples were analyzed. The correlation assay demonstrated good reproducibility within each group (Appendix A). The sequencing reads were analyzed with multiple pipelines to minimize the algorithm-dependent bias (more details in Methods). Eventually, we acquired a list of differentially expressed genes (DEGs) comprising 566 genes, with 42.9% (243/566) of DEGs upregulated and the rest, 57.1% (323/566), downregulated by MDA231-HEVs (Figure 6A, Appendix A). Prominently, the cell component enrichment analysis showed that 24.0% (136/566; 53 upregulated, 83 downregulated) of the DEGs were related to the extracellular EVs, which was the cause of these DEGs (Appendix A).

The gene ontology (GO) enrichment analysis of the DEGs revealed the pro-motility and pro-metabolism signatures of the DEGs. For the motility, the cell migration-related genes were enriched in both upregulated and downregulated DEGs, and the cell adhesion related genes were enriched in the downregulated DEGs (Appendix A). For the metabolism, several related GO terms were enriched in the upregulated DEGs (Appendix A). Consistently, the gene set enrichment analysis (GSEA) identified two upregulated hallmark gene sets: EMT (enrichment score [ES] = 0.475, *P* = 0.017) and oxidative phosphorylation (OP) (ES = 0.471, *P* = 0.004) (Figure 6B,C). These two gene sets included a total of 28 upregulated genes (Figure 6D,G), most of which were confirmed to be significant by RT-qPCR (Figure 6E,H). Importantly, we found that 89.3% (25/28) of these upregulated mRNAs were present in the MDA231-HEVs (Figure 6F,I), suggesting that the uptake of EV-mRNA at least partially contributed to the elevation of these genes’ expressions in HUVECs. Taken together, the cancer EVs carried EMT-related and OP-related mRNAs, and mediated changes in the recipient cell transcriptome that overrepresented cell motility and metabolism.

## 4. Discussion

EV cargo have been extensively investigated to discover biomarkers for cancer diagnosis. Among the EV cargo, proteins and miRNAs are the two major types of molecules that have been widely investigated. However, the feasibility of EV-mRNAs as biomarkers is rarely studied. Here, we quantified a panel of 61 hypoxia-targeted mRNAs in EVs and found that 30 of these mRNAs were present in cancer EVs and absent in normal EVs. Therefore, we propose that the previously unappreciated EV-mRNA is a promising type of cancer biomarker. A recent leading type of biomarker is the cell-free DNA (cfDNA), which takes advantage of the advances in nucleic acid quantification technologies [45]. Nevertheless, using cfDNAs as biomarkers can be problematic because of the controversial origin of cfDNAs and the exposure to DNase which can easily degrade the cfDNAs. As a result, the level of cfDNAs may unpredictably fluctuate, rendering this information less reproducible. In contrast, EVs were actively released by cells through endosomal pathways, with the cargo well-protected from degradation by enclosed membranes [41]. Notably, we found the EV-mRNA levels tended to be stable, since the coefficients of variance of independent replicates were all small (<0.3). Other works also showed that the profile of EV-mRNAs can reflect the physiological status of original cells [17]. Therefore, as EV-mRNAs can be quantified using similar technologies to that for cfDNAs, we envisage that EV-mRNA is a competitive alternative to cfDNA, because the actively secreted and membrane-enclosed EV-mRNAs are thought to be more informative, stable and abundant.

Among the assessed mRNAs that were present in cancer-derived EVs and absent in MCF10A-derived EVs, several genes, including SNAIL1/2, TAZ, AXL, RHOA and ROCK1, are well known to promote cell motility. SNAIL1/2 and TAZ are transcriptional factors or co-activators that induce EMT by suppressing epithelial markers like E-cadherin and upregulating mesenchymal markers such as vimentin [46]. The AXL receptor tyrosine kinase was implicated in an exacerbating feedback loop with SNAIL1/2, in which the overexpression of AXL elevated the SNAIL1/2 expression, and vice versa [47,48]. The RHOA/ROCK signaling axis could mediate the formation of actin stress fibers and focal adhesions, which could enhance the lamellipodium-based cell migration [49]. Taken together, these pro-migratory mRNAs might have synergized with the EV-proteins such as TGFβ1 [50], as well as miRNAs like miR-21 [51], to educate the recipient cell towards a migratory phenotype. Consistently, we observed an upregulation of the EMT gene set in HUVECs by MDA-MB-231-derived EVs. The activation of EMT in HUVECs can further contribute to angiogenesis [52], which is important for tumor progression.

The integrity of the EV-mRNAs remains controversial. While the length of intact mammalian mRNAs ranges from 400 to 2000 nucleotides (nt), that of exosomal mRNAs secreted by normal and cancer cells is distributed between 25 and 700 nt [53,54], indicating that full-length mRNA may be far less abundant than short-length RNAs in EVs. In fact, RNA size profiling by Agilent bioanalyzer showed that short-length RNAs (<50 nt) were mainly enriched in EVs [1]. In another study, 68.5% of EV-mRNAs detected by microarray were found to be in fragments, with 687 transcripts secreted from cells via exosome [54]. A total of 422 and 564 mRNA transcripts were detected by the nCounter platform in EVs from the plasma of cancer patients [55] and prostate cancer cells with mesenchymal phenotypes [56], respectively. Nevertheless, studies have found several full-length translatable mRNAs present in EVs. For example, mice cell MC/9-derived EV-mRNAs were translated into proteins with an in vitro rabbit lysate translation kit [1]. In addition, the EVs secreted by glioblastoma cells transduced with Gluc transfer Gluc mRNA to HBMVEC cells and induced a continuous increase in Gluc activity over 24 h [41]. We also observed that MDA231 cells transfected with GFP-tagged VEGFA secreted the EVs that carried GFP mRNA. The GFP-tagged VEGFA were transferred to HUVECs and translated to fluorescent GFP protein in 24 h. Taken together, the translatable mRNAs are present in EVs and can produce noticeable levels of proteins in recipient cells regardless of their relatively low abundance in EVs, and thus contribute to altering the recipient cells.

Interestingly, we observed a reduced proliferation rate of HUVECs co-cultured with the MDA231-HEVs. While the effect of tumor-derived EVs on promoting cell proliferation and escape from apoptosis has been extensively studied using immune cells or epithelial cells as target cells [57], the direct role of EV-mRNAs in affecting endothelial cells is not well known. As we found that the mRNAs of the two cyclin-dependent kinase inhibitors, CDKN1A and CDKN1B, were upregulated in HUVECs and that they were carried by the MDA231-HEVs, further investigation into whether they would be internalized and translated into proteins by the recipient HUVECs would be of particular interest to suggest their functional role in cell cycle regulation and cellular proliferation.

With the quantitative characterization of the EVs, we found the EV generation rate (EGR) was cell line-dependent. The non-tumorigenic MCF10A had a lower EGR than the cancer cells, which is consistent with previous works showing the level of EVs was enhanced in cancer patients [15]. Interestingly, the two cancer cell lines also had distinguishable EGR, with the highly motile MDA231 showing higher EGR than the less motile MCF7. Consistently, a previous study demonstrated that the inhibition of invadopodia formation reduced EV secretion [58]. Thus, we speculate that EGR is likely correlated with cell motility. The association of EV generation and cell migration can be further supported by a cell migration theory—the membrane flow model—whereby exocytosis, endocytosis and vesicle trafficking constitute a cycle of rearward membrane flow that accompanies the cell migration [59,60]. The speed of the membrane flow, which is associated with the frequency of exocytosis, was found to be higher in the cells migrating with faster velocity [60]. In addition, the migrating cells indeed deposited EVs on the path of migration [61]. Therefore, the cellular activities including EV generation, exocytosis, membrane flow and cell migration are likely to be associated with each other.

The increased frequency of EV secretion by migratory cancer cells may serve to elevate the level of circulating EVs in cancer patients, supplementing the previous understanding that the increased secretion of EVs is induced by the stresses in the tumor microenvironment, such as hypoxia [62]. Furthermore, we observed that MDA231 cells not only secreted more EVs, but also produced larger EVs than MCF10A and MCF7 cells. The generation of large EVs and control of plasma membrane rigidity depend on many different factors, including lipid or protein composition, cytoskeleton dynamics, cell metabolism, cytoplasmic viscosity, ions balance and signaling [63]. As these breast cell lines differ in terms of their membrane components [64] and morphological and structural properties with diverse mechanical characteristics [65], the observed size differences might be attributed to distinct processes related to EV biogenesis. Particularly, a specific class of large EVs, termed large oncosomes (LO), is increasingly being recognized as a potential diagnostic/prognostic cancer biomarker, with their functional roles in cancer development and progression [63]. While their relationship with the length of mRNAs is not known, a previous study found a higher abundance of exosomal proteins in MDA231 cells compared to MCF7 cells [66]. Altogether, these results indicate that besides EV cargoes, the biophysical features of EVs may also be informative for cancer diagnosis. Additionally, we found that remarkably more types of mRNAs were present in cancer EVs than normal EVs, with higher loading frequency.

Hypoxia as a typical stress in the TME can induce the formation of new blood vessels, i.e., angiogenesis, which in turn boosts the supply of oxygen to mitigate the hypoxic stress. This feedback response is largely mediated by the key protein VEGFA. In this study, we found that a high level of VEGFA mRNA was carried by EVs derived from multiple types of cancer, and the level was modulated by the hypoxic stimulus. Furthermore, treating HUVECs with the VEGFA mRNA^+^ EVs promoted angiogenesis via a VEGFR-dependent pathway. These results indicate that cancer cells may favor angiogenesis not only through secretory VEGFA protein, but also by transferring VEGFA mRNAs via EVs.

The study of hypoxia-induced cancer EVs is still at a stage of infancy. Evidence suggests that hypoxia-induced cancer EVs can educate the cells in the TME and induce angiogenesis, invasion, metastasis and immune evasion, with these effects mainly attributed to certain proteins and miRNAs harbored by EVs [19]. In line with these prior works, we observed that hypoxia-induced cancer EVs promoted the VEGFR-dependent angiogenesis and elevated the expression of EMT-related genes in HUVECs. Rather than proteins or non-coding RNAs, the identification of angiogenesis-related mRNAs, such as VEGFA, and EMT-related mRNAs, such as SNAIL1/2, in the EVs indicates that the mRNAs are at least partially implicated in reprograming the recipient cells. Interestingly, we found that with hypoxic stress, most of the hypoxia-targeted mRNAs were downregulated in MCF7-EVs but upregulated in MDA231-EVs. This contradictory response to hypoxia could not be adequately explained by the HIF-dependent transactivation of genes, which always upregulates the target genes. Future investigations of the machinery that packs mRNAs into EVs could extend our understanding of how hypoxia regulates the EV-mRNAs, thus opening up potentially new avenues for the diagnosis and treatment of cancer.

## 5. Conclusions

Overall, we show that the transcripts of hypoxia-targeted oncogenes, EMT-related genes and metabolism-related genes prevail in cancer EVs. The hypoxic stress in the TME can induce the loading of more types of mRNAs into EVs and regulate the abundance of EV-mRNAs in a cell line-dependent manner. The EV-mRNAs are likely functional since they can be internalized and translated into proteins by the recipient HUVECs. Accompanied by the uptake of these EV-mRNAs, the HUVECs were altered toward an angiogenic phenotype with the upregulation of EMT-related and metabolism-related genes (Figure 7).

## Figures and Tables

**Figure 1 cancers-13-02009-f001:**
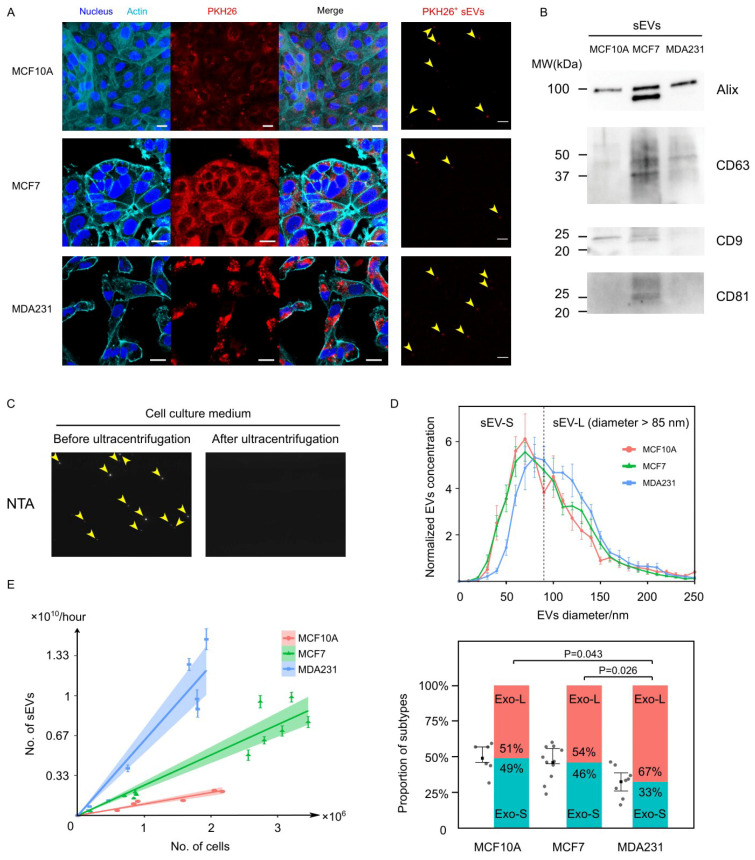
Characterization of cancer and normal cell-derived EVs. (**A**) Breast cell lines MCF10A, MCF7 and MDA231 were stained with cell membrane dye PKH26 and incubated to generate PKH26+ EVs. Scale bars = 20 µm for left three columns and scale bars = 5 µm for the sEVs column. (**B**) An estimated 10 ug of total sEV proteins were used for western blot. The sEV protein markers including Alix, CD63, CD9 and CD81 were enriched. Because the BCA protein assay had limited accuracy at the range of EV protein concentration, the actual amount of protein in each lane might vary. (**C**) Crude EVs in the culture medium were detected and counted by Nanoparticle Tracking Analysis. Ultracentrifugation removed all the detected particles, suggesting that these particles were indeed EVs. (**D**) Normalized size distribution of EV for MCF10A (*N* = 6), MCF7 (*N* = 11) and MDA231 (*N* = 8). The EVs derived from all three cell lines have a diameter of 50–150 nm, with the MDA231-derived EVs showing a rightward shift compared to the other two cell lines. Dividing the EVs into smaller and larger subtypes with a threshold diameter of 85 nm, the MDA231-sEVs constitute prominently higher proportions of sEV-L subtype. Error bars stand for the 25th and 75th percentile. *P* values are calculated based on Mann–Whitney U test. (**E**) Scatter plot shows EV number and cell number were linearly correlated, with the coefficients distinguishable between cell lines. The rank of the EV generation rate was MCF10A < MCF7 < MDA231. Error bars of each point stand for the standard error of the mean (SEM). Data were fitted by linear regression with zero intercept, with the 95% confidence interval highlighted in color.

**Figure 2 cancers-13-02009-f002:**
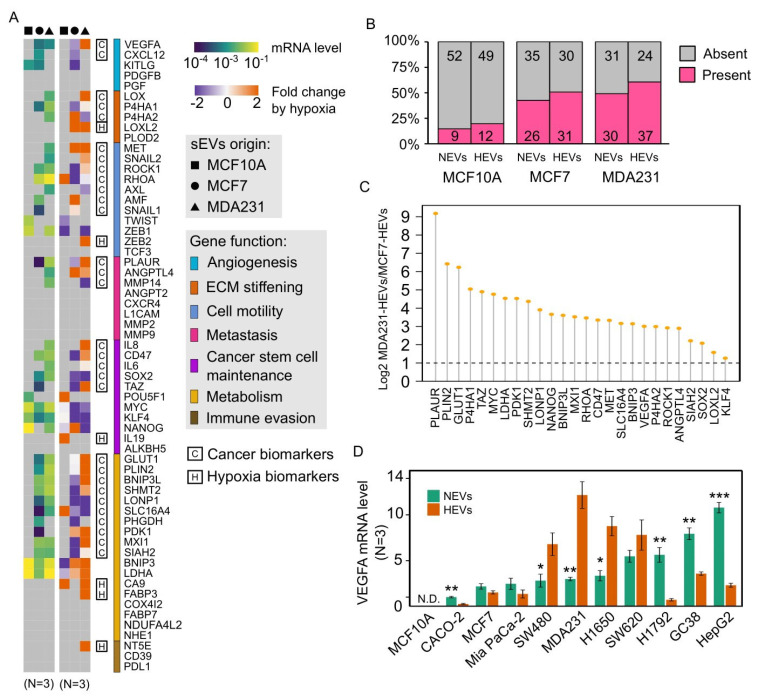
Quantification of EV-mRNAs of the hypoxia gene panel. (**A**) The two heatmaps show the normoxic EV-mRNA levels (left) and the hypoxia-induced log2 fold changes of EV-mRNA level (right). The expression level is averaged from three independent experiments. The 61 genes are grouped by their contributions in cancer development. Potential cancer biomarker and hypoxia biomarker genes are marked. (**B**) The proportion of the hypoxia gene panel whose transcripts were detected in the EVs. The two cancer cell lines-derived EVs harbored the mRNAs of a much larger proportion of the hypoxia gene panel. (**C**) MCF7-HEVs and MDA231-HEVs carried 26 mRNAs in common from the panel. All these genes had > 2 folds higher mRNA levels in MDA231-HEVs compared to MCF7-HEVs. (**D**) VEGFA mRNAs were present in the EVs derived from multiple cancer cell lines with varied levels. Most of the EV VEGFA mRNA levels were significantly altered by hypoxia (*N* = 3). * *P* < 0.05, ** *P* < 0.01, *** *P* < 0.001 based on Student’s *t* test for (**D**).

**Figure 3 cancers-13-02009-f003:**
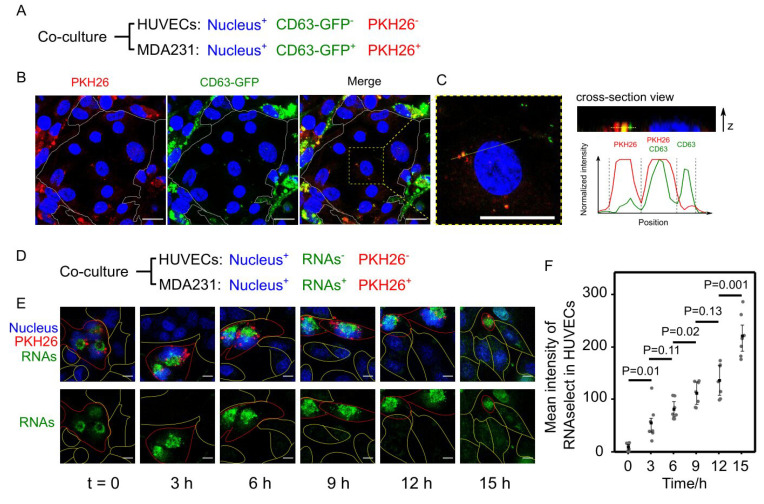
Fluorescent trafficking of the transfer of EVs and RNAs from MDA231 to HUVECs. (**A**) HUVECs and MDA231 were differentially stained and co-cultured. CD63-GFP and PKH26 were used to double label the MDA231-derived EVs. (**B**) After 24-h co-culture, EVs labeled by CD63-GFP or PKH26 were internalized by HUVECs. Scale bars = 20 µm. (**C**) Enlarged view of internalized EVs in HUVECs. The z-position of the EVs did not exceed the height of nucleus, indicating that the EVs were inside the cell rather than on the cell surface. The EVs were either PKH26+ or CD63+. Scale bars = 20 µm. (**D**) SYTO RNASelect was used to stain RNAs in MDA231. (**E**) The fluorescent RNAs from MDA231 gradually accumulated in HUVECs in 15-h live cell imaging. MDA231 and HUVECs are marked by lines in red and yellow, respectively. Scale bars = 10 µm. (**F**) Significant elevations of fluorescent RNAs in HUVECs were observed after 3 h (*N* = 7). The average intensity of RNAs in HUVECs was measured by ImageJ and the *P* value was calculated based on Student’s *t* test.

**Figure 4 cancers-13-02009-f004:**
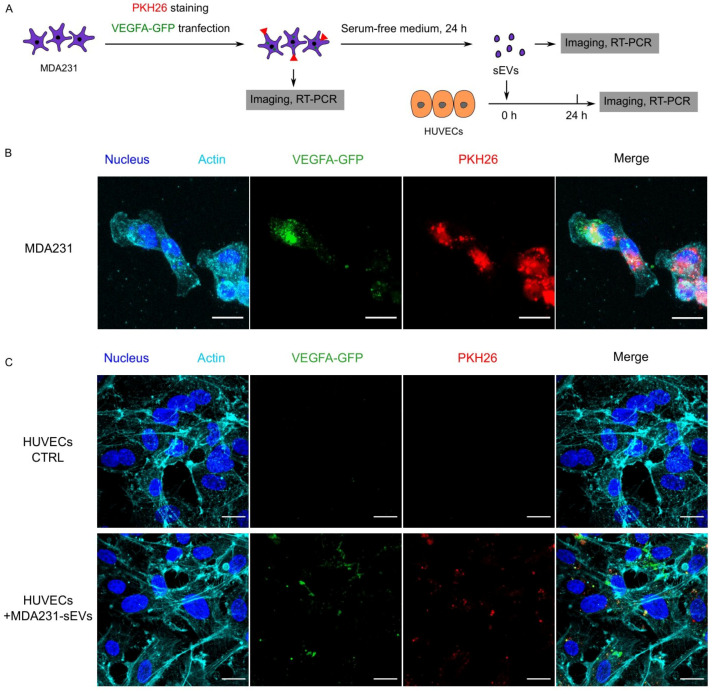
Internalization and translation of cancer EV-delivered VEGFA-GFP mRNA by HUVECs. (**A**) MDA231 cells were stained with PKH26 and transfected with VEGFA-GFP, as shown in (**B**). This stained MDA231 was then incubated in serum-free culture medium to generate EVs. These EVs (50 µg/mL) were used to treat HUVECs for 24 h. (**C**) After 24-h co-culture with the PKH26^+^ VEGFA-GFP mRNA^+^ EVs derived from (**B**), HUVECs internalized the PKH26^+^ EVs and synthesized VEGFA-GFP protein from the EV-mRNA. Scale bars = 20 µm for (**B**,**C**). (**D**) No detectable level of VEGFA-GFP protein was harbored by the PKH26^+^ EVs derived from (**B**). Scale bars = 5 µm for (**D**). (**E**) DNA gel electrophoresis showed that the VEGFA-GFP^+^ MDA231 transferred VEGFA-GFP mRNAs to HUVECs through EVs.

**Figure 5 cancers-13-02009-f005:**
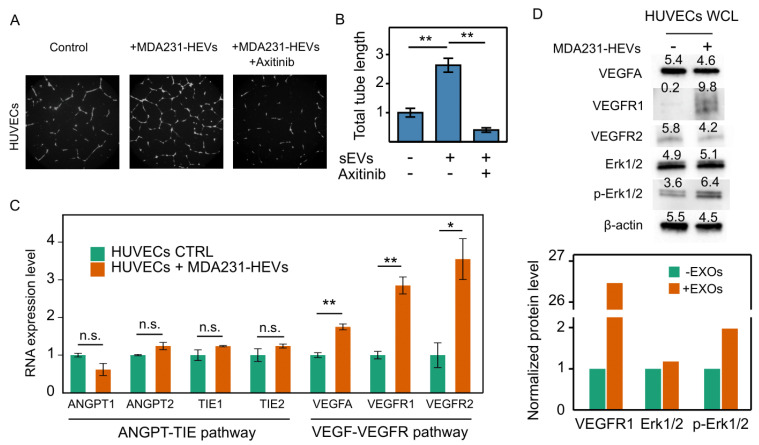
Downstream effects of cancer EVs on angiogenesis. (**A**) Tube formation assay of HUVECs incubated in basal medium (control), treated with 50 µg/mL MDA231-HEVs, or treated with both MDA231-HEVs and 0.1 µM Axitinib (inhibitor of VEGFR) for 24 h. The EVs induced formation of denser tube networks, which can be efficaciously blocked by Axitinib. Images are representative of two independent experiments. (**B**) The total tube length of images from the same experiment were analyzed by ImageJ and normalized by the average of the control group (*N* = 6). (**C**) MDA231-HEVs upregulated the VEGF-VEGFR angiogenesis pathway related gene, while showing no effect on the other ANGPT-TIE angiogenesis pathway in HUVECs (*N* = 3). (**D**) Western blot of proteins involved in the VEGF-VEGFR angiogenesis pathway. MDA231-HEVs conspicuously enhanced the level of VEGFR1 and the phosphorylation of downstream Erk1/2. Average intensities of the bands were first normalized to β-actin, and then normalized to the control group. Images are representative of two independent experiments. n.s.: not significant, * *P* < 0.05, ** *P* < 0.01 based on Mann–Whitney U test for (**B**), or Student’s *t* test for (**C**). Error bars stand for SEM for (**B**,**C**).

**Figure 6 cancers-13-02009-f006:**
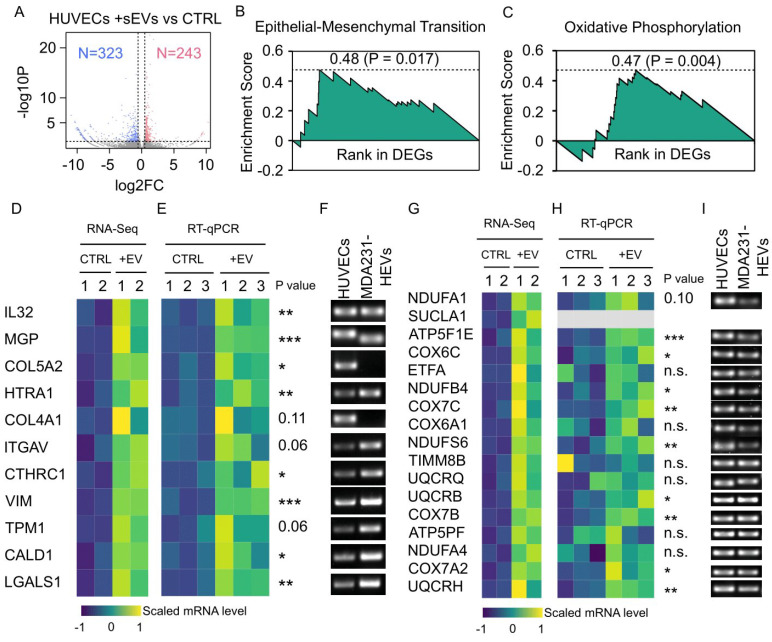
RNA-seq analysis of HUVECs treated with cancer EVs. (**A**) Volcano plot shows RNA-seq data of HUVECs with or without the treatment by 50 µg/mL MDA231-HEVs. The fold change and *P* value were analyzed by HISAT2-HTSeq-DESeq2 pipeline. Differentially expressed genes (DEGs) were highlighted in red for 242 upregulated genes and blue for 323 downregulated genes. Gene set enrichment analysis of the DEGs revealed that EMT gene set (**B**) and OP gene set (**C**) were significantly enriched, with enrichment scores of 0.475 (*P* = 0.017) and 0.471 (*P* = 0.004), respectively. (**D**) Heatmap shows the RNA-seq data of upregulated DEGs that were annotated by EMT gene set. (**E**) RT-qPCR verified the upregulation of the DEGs in (**D**). (**F**) DNA gel electrophoresis demonstrated that, except for COL5A2 and COL4A1, the mRNAs of all the DEGs in (**D**) (9/11) were present in MDA231-HEVs. (**G**) Heatmap shows the RNA-seq data of upregulated DEGs that were annotated by OP gene set. (**H**) RT-qPCR verified the upregulation of the DEGs in (**G**). (**I**) DNA gel electrophoresis demonstrated that, except for SUCLA1, the mRNAs of all the DEGs in (**G**) (15/16) were present in MDA231-HEVs. n.s.: not significant, * *P* < 0.05, ** *P* < 0.01, *** *P* < 0.001, based on Student’s *t* test for (**E**,**H**).

**Figure 7 cancers-13-02009-f007:**
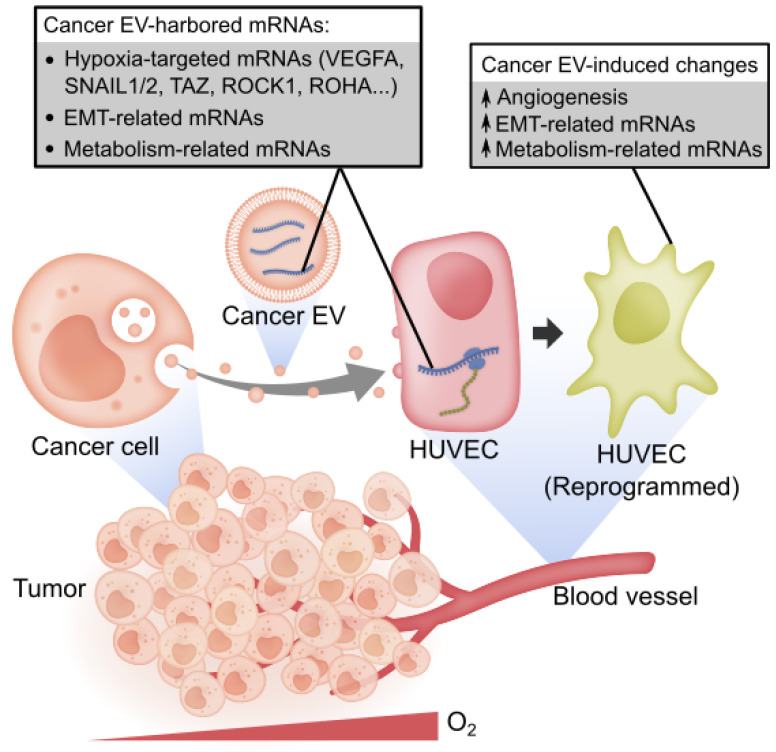
Summary model. Cancer cell-derived EVs carry diverse mRNAs including hypoxia-targeted mRNAs, EMT-related mRNAs and metabolism-related mRNAs. The EV-mRNA profile is not strongly correlated with cell mRNA profile. A proportion of these EV-mRNAs can be regulated by the hypoxic stress in the tumor microenvironment. Cancer EVs deliver translatable mRNAs, such as VEGFA, to HUVECs, which contribute to activating the HUVECs toward an angiogenic phenotype, accompanied by the upregulation of EMT-related and metabolism-related genes in HUVECs.

## Data Availability

The RNA-seq data generated in this study have been submitted to the National Center for Biotechnology Information GEO (GSE153694; reviewer token: opajsscwpdezpkx). All data needed to evaluate the conclusions in the paper are present in the paper and/or the Appendix A. Additional data or materials related to this paper may be requested from the authors.

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
