# Peer review of "Distinct mRNAs in Cancer Extracellular Vesicles Activate Angiogenesis and Alter Transcriptome of Vascular Endothelial Cells"

_cancers, 2021, doi:10.3390/cancers13092009_

Round 1
Reviewer 1 Report
The article is devoted to the study of the potential of tumor cell-produced vesicles, as biomarkers of the cancer progression. The authors provide a comprehensive analysis of the mRNA profile of small vesicles isolated from breast cancer cell lines under normoxic and hypoxic conditions. However, the authors need to address the next comments before the manuscript can be accepted for publication.
1 There should be not any results of the research in the Introduction section. It is better to describe in more detail why vesicles can become more efficient biomarkers compared to other cancer biomarkers existing today.
2 Page 3, Line 98. It is unclear how hypoxia was carried out?
3 Page 3, Line 108. Was EV stability at -80°C without a cryoprotectant checked somehow? How long the samples were stored at -80°C before the use?
4 You should add a subsection dedicated to statical analysis into the Materials and methods.
5 It is not clear why the authors decided to choose breast cancer cells for their research. Could you explain it?
6 Figure 1. Scale bars should be added in the pictures.
7 Figure 1. Such a magnification and confocal microscopy do not allow observing the structure of harvested EVs. According MISEV guidelines analysis of single vesicles is also required, in particular TEM or SEM to analyze EV morphology.
8 Figure 1. You should specify how much protein has been loaded for Western blot analysis.
9 Figure 4. Scale bars are required.
10 Figure 5. It is mentioned that the data are obtained by two independent experiments. At the same time in the tube formation assay 6 replicates were used for the statistical analysis (n ​​= 6). It is not clear, are these technical replicates obtained from two independent experiments? It is necessary to explaining more clearly how replicates were obtained and indicate it in the Figure legends.
11 How the authors can explain why the size of the vesicles harvested from the MDA-MB-231 cells is higher than the size of the vesicles harvested from other cell types? And, in their opinion, if the size can affect the transfer of mRNAs, maybe it can correlate with the length of mRNAs packaged into EVs?
12 The authors did not discussed properly the interesting fact that cultivation with EVs leads to a decrease in HUVEC proliferation and how it can be explained.
Reviewer 2 Report
Distinct mRNAs in cancer extracellular vesicles activate angiogenesis and alter transcriptome of vascular endothelial cells
In this study, Zhang et al. sought to understand the role of hypoxia-induced mRNA changes in cancer-derived small extracellular vesicles (sEVs), and how this can alter recipient cells, in this case HUVECs. Overall, there is an impressive amount of work with some interesting insights and well-designed experiments. In general, the manuscript is well written and presented. Although this is an interesting piece of work, there are some concerns that need to be addressed before this article should be accepted.
General comments:
Initially, some of the references used are somewhat outdated, although still important. I would suggest to also include more recent reviews that support the use of the more updated terminology such as ‘sEVs” and that detail the contents of these vesicles.
The comment on line 75 that VEGFA containing sEVs could be a pan-cancer biomarker is an overstatement, particular with the use of a single ‘normal’ cell line, and the fact several cancer lines showed reduced VEGFA mRNA in response to hypoxia.
Line 295 – given that the work characterised breast cancer sEV secretion, the claim cancer cells secrete more EVs than healthy cells should be rephrased to “breast cancer cells”.
Please include size-bars on all fluorescent microscopy images.
Cell culture-were the cells STR profiled or tested for mycoplasma?
In regard to line 256 -258 “potential contaminant that can mimic sEVs is very low density lipoprotein (VLDL) since the VLDL could have a similar size around 100nm. But sEVs (1.1-1.2 g/mL) and VLDL (~1.006 g/mL) can be differentiated by centrifugation.” Is there any reason to suspect VLDL vesicles are secreted from the cell lines used in this study?
Given that this study uses hypoxic conditions to elicit a phenotype, there should be some characterisation of HIF stabilisation. That is, to determine a hypoxic response has been elicited, you need to show either HIF1/2 stabilisation under the oxygen conditions used. This is particularly important as some cell lines do not seem to respond the same way with VEGFA content for example.
Figure 1 The use of “Exo L”, or “sEV-L” etc, as you define your vesicles as sEVs, this needs to be altered.
Figure 1a – size bar in image of cells, the fluorescence in the media is not convincing of EVs, could be large aggregates or membrane fragments
Figure 1b western blot – was this the same protein loaded? Because ‘canonical’ EV proteins differ greatly. Importantly. It is required to add negative markers like calnexin or GM130 as per MISEV guidelines.
Figure 2 Are the EV mRNA levels on left panel of the heatmap from normoxic conditions?
Figure 4 the evidence for VEGFA-GFP mRNA and not VEGFA-GFP protein in sEVs is not convincing. Another method, such as nanoparticle analysis with fluorescence (Nanosight/Zetaview/nanoFCM), or western/ELISA should be used to confirm this, otherwise the statement that the mRNA is responsible is not sufficiently supported.
Justifying using 50 μg/mL with a single article that shows a range of 500 - 1500 μg/mL in cancer patients is not sufficient. Given patient sEVs have different contaminants, and are also composed of sEVs from multiple tissues, this is not a good proxy for in vitro-derived sEVs from a single source. More importantly, what was the concentration (particles/mL) used in this study.
Reviewer 3 Report
In this article, Zhang and colleagues, focus their attention on the potential of EV mRNA profile to reflect the cell malignancy and the intercellular transfer of these mRNAs to contribute toward tumor angiogenesis. The study was well conducted and the results are clearly demonstrated. Also, the methodology section is well described and doesn't need particular revision. A point that could be improved is related to the introductive part of the paper addressing the need for new technologies for the association of a specific marker with an exosome subtype and the exosome subtype to a particular function and/or group of functions s(PMID: 32759810 is just an example).
Reviewer 4 Report
This is an interesting manuscript. Although many papers showed functions of miRNAs and proteins in extracellular vesicles (EVs), very little papers showed mRNA functions in EVs. In this manuscript, Zhang et al. focused on mRNA in EVs derived from cancer cells. Firstly, the authors characterized cancer EVs by quantifying EV numbers among cell lines and separating them based on the size. Secondly, gene expression of hypoxia-related genes examined, and cancer cell lines expressed them more than MCF10A, normal epithelial cell line. Thirdly, cancer EVs were shown to be transferred into HUVEC, and then mRNA (VEGFA-GFP) in cancer EVs were translated into the protein in the HUVEC. Finally, using RNA-seq, the effect of cancer EVs in HUVEC was investigated. Overall, the manuscript was clearly written and well-organized; however, several concerns need to be addressed before considering publication. Specific comments are shown below.
Major comments:
- As the authors also a bit mentioned in the discussion, how many intact mRNAs can be packaged into an EV? In the case of miRNAs and proteins, miRNA is very small and stable, so it can be packaged into EVs, and also some membrane protein locates on the EV membrane. When considering the size of mRNA (a few thousand bp) and its stability, I imagine that not so many mRNAs can be packaged into the EV. Is it theoretically calculatable? Or at least, it is better to discuss in the manuscript.
- In Figure 3, the authors used a co-culture system to show the EV transfer from cancer cells to HUVEC. When cells attached together, other systems such as tunneling nanotubes could transfer proteins and mRNAs (https://www.frontiersin.org/articles/10.3389/fmolb.2017.00050/full). The authors should use a transwell system to demonstrate the mRNA transfer via EVs.
- In Figure 4D, the authors show no VEGFA-GFP proteins in cancer EVs. Why this protein is not packaged?
- When VEGFA mRNA in the EVs is taken up by HUVEC, what happens? Does HUVEC secrete VEGF into extracellular space, and does the VEGF work to HUVEC in an autocrine manner?
- In Figure 5C and D, mRNA of VEGFA increased in HUVEC + MDA231-HEVs but the protein of VEGFA was slightly decreased. Is there any explanation for the discrepancy?
Minor comments:
- In Figure 6F, why are the band sizes of ATP5F1E different between HUVECs and MDA231-HEVs?
Round 2
Reviewer 4 Report
The authors fully addressed my original concerns.